# OpenReview forum: "EMBGuard: Constructing Hazard-Aware Guardrails for Safe Planning in Embodied Agents"
_ICML.cc/2026/Conference — ICML 2026 regular_

### Official Review · Reviewer_5mrG · 2026-03-13

**Soundness:** 3
**Presentation:** 3
**Significance:** 3
**Originality:** 3
**Overall Recommendation:** 3
**Confidence:** 3

**Summary:**

This paper benchmarks physical risk assessment ability for MLLMs. It evaluates whether a candidate action is risky given a scene image. It constructs a 17K gemini-3-pro generated training set and a 189 real-world test set for evaluation.

**Compliance With Llm Reviewing Policy:**

Affirmed.

**Final Justification:**

The rebuttal addressed part of my concerns but not the main one. I maintain my original score.

**Key Questions For Authors:**

See weakness section.

**Limitations:**

No discussion of limitations. Please try to solve or discuss the limitations mentioned in weakness section.

**Strengths And Weaknesses:**

Strengths:

1. Well-motivated problem with clear formulation; compositional variation systematically covers hazard-action combinations that prior safety benchmarks lack.
2. Compact 2B/4B models achieve competitive performance with proprietary MLLMs, demonstrating targeted training can substitute scale for this task.

Weaknesses:

1. The test set with real images has only 189 samples across 7 risk types × 4 scenario types, making per-category analysis statistically underpowered.
2. The hazard accuracy relies on GPT-4o as judge, but no quantitative human-judge agreement analysis is provided.
3. Each (image, action) pair is judged independently with no temporal context, so sequential risks from prior actions might be missed. For example, "place towel on counter" is safe alone but risky if the agent previously turned on a nearby stove. This also reflects a broader issue of ignoring partial observability, which is common in embodied agent settings.
4. Actions are text-level descriptions, limiting applicability to high-level planners; continuous-control VLAs cannot use this guardrail directly.

---

> ### Author Rebuttal · Authors · 2026-03-31
>
> # **W1: Limited Real-World Image Dataset Size**
> We thank the reviewer for pointing out that the dataset size may make the per-category analysis statistically underpowered. We also acknowledge that the original EMBGuardTest dataset was limited in size. Therefore, after submission, we additionally created more instances following the same guidelines used to manually construct the original dataset. As a result, we added 140 instances that do not overlap with the existing training and test sets, bringing the total to 329 instances.
>
> **Note that we re-ran all experiments with the updated EMBGuardTest, and while the numbers have slightly changed, the core findings from the original paper remain consistent.** To support this, we report the updated main performance (original Table 2\) and dataset correlation (original Figure 5\) results below. We will reflect all of these updates in the camera-ready version.
>
> ### **Due to the character limit, we present the table at "Initial Response to Reviewer a77M W1 Response"**
> # **W2: LLM-as-a-Judge approach on Hazard accuracy**
>
> Thank you for identifying the gap we had left unaddressed. To provide quantitative human judge agreement analysis, we conduct a study comparing the results of the LLM judge and human judges. We sample 100 Risky Scenarios with gold hazard labels, balanced across the four source models (GPT-5.1, Qwen-3-VL-235B-a22b, EMBGuard-2B, EMBGuard-4B) and GPT-4o's judgment outcomes (correct/incorrect, 50/50). Note that this size yields a margin of error of ±0.10 for Cohen's κ at the 95% confidence level. Three CS undergraduate students independently annotate each sampled instance, with final labels determined via majority voting. Cohen's Kappa (κ) between majority-voted human labels and GPT-4o's judgments is then computed.
> |Student| Accuracy (%) |
> |-|-|
> |1|94%|
> |2|84%|
> |3|86%|
> |Average (Majority Voted)|95%|
>
> ||Cohen's Kappa (κ)|
> |-|-|
> |GPT-4o \- Human (Majority Voted)|0.90|
>
> The high agreement (κ \= 0.90) validates GPT-4o as a reliable judge in our setting. We will include the human judge agreement results in Appendix B.1.
>
> # **W3: Temporal context awareness in EMBGuard**
> We understand your concern that our setting may not fully account for temporal context. We note, however, that **EMBGuard takes the current scene observation as input rather than a textual description, which allows it to visually capture the consequences of prior actions.** For example, a stove that was previously turned on would appear as a lit stove in the current observation, and thus be directly detectable. This means EMBGuard can assess physical risk without relying on sequential action history, unlike the agent policy itself which must track prior actions to reason about the current state.
>
> However, we acknowledge that this assumes the scene input fed into the guardrail captures all relevant hazardous conditions. In real-world scenarios, there may be cases where the robot's visual sensor fails to capture certain risks, such as a lit stove nearby. We will therefore add the following to the Limitation section.
> > **Visual Sensor Coverage Assumption.** Our guardrail assumes that the scene observation provided as input fully captures all relevant hazardous conditions present in the environment. However, in real-world deployments, a robot's visual sensor may fail to capture certain risks due to limited field of view, occlusions, or sensor noise. As a result, hazards that are not visually reflected in the observation — such as a lit stove outside the camera's field of view — may go undetected by the guardrail. We recognize this as an inherent limitation of our current setup and highlight it as an important direction for future work.
>
> # **W4: Limited applicability for text-based action**
> We agree with your concern regarding the limited applicability of our guardrail to text-based action descriptions. While our guardrail operates on text-level action descriptions, it does not directly apply to continuous-control policies such as VLAs. Extending safety reasoning to such settings remains an important direction for future work, and we hope our dataset and the core philosophy of decoupling safety reasoning from task execution can serve as a foundation for this. We will add the following to the Limitation Section, as you suggested.
> > **Applicability to Continuous-Control Policies.** Our current guardrail is designed to operate on text-level action descriptions and does not directly extend to continuous-control policies such as Vision-Language-Action models (VLAs). While our framework effectively decouples safety reasoning from task execution at the semantic level, applying this philosophy to low-level motor control remains a non-trivial challenge. We recognize this as a limitation of our current work and highlight extending safety reasoning to continuous-control settings as an important direction for future research.

---

> > ### Author Rebuttal · Reviewer_5mrG · 2026-04-03
> >
> > W1 and W2 are addressed. The expanded test set and the human-judge agreement study resolve my original concerns.
> > W3 remains partially resolved. The authors' argument that visual observation captures consequences of prior actions (e.g., a lit stove is visible) is reasonable for some cases, but does not fully address the broader partial observability issue, which is common in embodied scenarios.
> > W4: I thank the authors for the promise of adding a limitations section.

---

> > > ### Author Response · Authors · 2026-04-06
> > >
> > > ## 2nd Response to Reviewer 5mrG on W3:
> > > Thank you for your follow-up. **We would like to first discuss the rationale behind the design choice of our guardrail model, and then also share our thoughts on your concern.**
> > >
> > > The most important consideration in designing our guardrail model was real-time deployability. To allow EMBGuard to operate without imposing significant overhead on execution, we therefore designed it to assess action risk on a per-step observation-action basis. As we discussed earlier, our hypothesis was that as EMBGuard takes the current scene observation as image input, it will visually capture the consequences of prior actions. And **we still believe that this design would be capable of handling the vast majority of risks in practice.**
> > >
> > > **However, we recognize that this per-step design could omit some cases.** First, in embodied agents’ POMDP setups, EMBGuard can potentially not detect **risks where there is a mismatch on the belief state of the embodied agent and the actual state, and when the risk becomes apparent over a long horizon**. Second, issues may arise from the inherent ambiguity of real-world objects, where **the identity or properties of an object may be uncertain at the time of action, making it difficult to assess risk without additional context or clarification**.
> > >
> > > **We agree that this is an important direction for future work.** A natural next step would be to extend EMBGuard to accept additional context from the agent, such as action history, task state, or additional information about the object, enabling more comprehensive risk assessment while preserving the modular design of our framework.
> > >
> > > **We will also add the following to the Limitation section of the revised paper:**
> > > > **Partial Observability.** Our guardrail evaluates each observation-action pair independently, assuming that the consequences of prior actions are visually reflected in the current observation. However, this assumption can break down in real-world settings — for instance, when there is a mismatch between the agent’s belief state and the actual environment, when risks only emerge over a long horizon, or when the identity or properties of an object are inherently ambiguous. We highlight extending EMBGuard to incorporate broader context, such as action history or richer object-level information, as an important direction for future work.

---

### Official Review · Reviewer_a77M · 2026-03-13

**Soundness:** 2
**Presentation:** 3
**Significance:** 3
**Originality:** 3
**Overall Recommendation:** 4
**Confidence:** 3

**Summary:**

This paper proposes EMBGUARD, a multimodal guardrail for embodied agents that detects action-conditioned physical risks from image observations and candidate actions. The model predicts whether an action is risky, identifies the risk category, and provides a textual explanation of the hazardous configuration. To support this task, the authors introduce two datasets: EMBHAZARD (17K synthetic training samples) and EMBGUARDTEST (189 real-world evaluation scenarios). Experiments show that the proposed model achieves competitive performance compared with larger MLLMs and reduces the over-conservative bias observed in general-purpose models. This work appears to analyze a relevant issue, and overall, the article focuses on a relevant question regarding safety mechanisms for embodied agents operating in real-world environments.

**Compliance With Llm Reviewing Policy:**

Affirmed.

**Final Justification:**

I appreciate the authors' effort, but the rebuttal does not change my judgment.

**Key Questions For Authors:**

1.How sensitive is the model performance to the specific image generation model used in the dataset construction pipeline?

2.Did the authors consider alternative dataset generation approaches, such as simulation-based data collection using robotics simulators?

**Limitations:**

yes

**Strengths And Weaknesses:**

Strengths

1.The paper addresses a critical problem in embodied AI: ensuring that agents can recognize potentially hazardous interactions between actions and environmental objects.

2.The dataset construction pipeline is one of the strongest components of the paper.The compositional design (causal risky, selective risky, decoupled benign, and absent benign scenarios) is particularly valuable because it forces models to distinguish between hazards that are actually triggered by actions and those that are not.The use of scene graphs to control hazard relationships is a thoughtful design choice that enables systematic variation while preserving causal structure.

3.The paper compares EMBGUARD with a wide range of baselines.Such comprehensive benchmarking strengthens the credibility of the results.

Weaknesses

Most of the training data is synthetically generated using image generation models. While the authors demonstrate correlation between synthetic and real-world performance, the real-world test set (189 images) is relatively small.

---

> ### Author Rebuttal · Authors · 2026-03-31
>
> ## Initial Response to Reviewer a77M
>
> We sincerely appreciate the reviewer for recognizing the strengths and contributions of our work. We will address the weakness and questions you have raised below.
>
> ---
>
> # W1 Synthetic Data and Real-World Dataset size Concern:
>
> We fully understand the concern regarding the limited size of EMBGuardTest, despite having demonstrated the correlation between the synthetic dataset (held-out set from EMBHazard) and the real-world dataset (EMBGuardTest) as you pointed out. Sharing this concern, after submission, we additionally created more instances following the same guidelines used to manually construct the original dataset. As a result, we added 140 instances that do not overlap with the existing training and test sets, bringing the total to 329 instances.
>
> **Note that we re-ran all experiments with the updated EMBGuardTest, and while the numbers have slightly changed, the core findings from the original paper remain consistent.** To support this, we report the updated main performance (original Table 2) and dataset correlation (original Figure 5) results below.
>
> Notably, the updated EMBGuardTest demonstrates a high correlation with the held-out set of EMBHazard, and we hope this mitigates your concern. We will reflect all of these updates in the camera-ready version.
>
> |Model Name|Potential Risk|Risk Type|Hazard|
> |---|---|---|---|
> |InternVL3.5-1B|16.2 (±2.6)|14.6 (±5.8)|9.8 (±3.0)|
> |InternVL3.5-2B|44.9 (±1.2)|21.1 (±4.4)|5.1 (±3.2)|
> |Qwen-3-VL-2B|47.2 (±0.7)|37.5 (±1.8)|5.9 (±2.5)|
> |Qwen-3-VL-4B|47.3 (±0.6)|51.0 (±0.0)|10.5 (±3.5)|
> |Gemma-3-4b|39.9 (±0.7)|27.1 (±4.5)|9.6 (±1.6)|
> |Qwen-3-VL-8B|49.1 (±0.7)|51.6 (±2.0)|14.4 (±2.3)|
> |Gemma-3-12b|47.3 (±0.9)|49.5 (±1.8)|9.2 (±2.7)|
> |Gemma-3-27b|45.3 (±0.5)|63.7 (±2.2)|16.4 (±4.6)|
> |Qwen3-vl-30b-a3b|46.1 (±1.6)|64.0 (±5.6)|24.8 (±5.2)|
> |Qwen-3-VL-32B|49.7 (±1.2)|56.9 (±0.4)|24.6 (±2.6)|
> |Qwen3-vl-235b-a22b|49.5 (±1.0)|56.4 (±2.9)|26.7 (±1.3)|
> |GPT-4o-mini|51.1 (±1.0)|52.3 (±2.4)|23.6 (±3.5)|
> |GPT-4o|52.3 (±1.8)|51.8 (±1.3)|28.8 (±4.2)|
> |GPT-5.1|55.8 (±2.7)|58.1 (±1.2)|33.4 (±4.4)|
> |Gemini-2.5-Flash|56.8 (±1.5)|55.5 (±2.4)|27.0 (±2.2)|
> |Gemini-2.5-Pro|58.4 (±1.2)|56.8 (±1.1)|29.3 (±4.2)|
> |**EMBGuard-2B**|51.6 (±1.1)|44.6 (±3.3)|7.4 (±3.1)|
> |**EMBGuard-4B**|54.3 (±1.7)|50.3 (±0.8)|14.6 (±1.5)|
>
> Table. Performance comparison of safety guardrail models.
>
> |Metric|Correlation (r)|P-value|
> |---|---|---|
> |Potential Risk|0.960|<0.001|
> |Risk Type|0.883|<0.001|
> |Hazard|0.813|<0.001|
>
> Table. Dataset correlation (Pearson).
>
> # Q1: Model Sensitivity to Image Generator
>
> Thank you for raising this question. Initially, we used gemini-2.5-flash-image, but found that it struggled to preserve critical spatial relationships and core hazard configurations as scene complexity increased, with limited photorealism as well. We therefore switched to gemini-3-pro-image-preview, which was the state-of-the-art at the time and successfully addressed these issues. Since image generation quality directly impacts guardrail model performance — which fundamentally relies on spatial relationships between objects and their physical states — we treated the choice of image generation model as an important design decision in our pipeline. **Therefore, based on our experience, the choice of image generation model significantly affects image quality and photorealism, and our experimental results suggest that gemini-3-pro-image-preview model is sufficiently effective for generating synthetic data in our pipeline.**
>
> # Q2: Alternative Dataset Generation Methods
> We appreciate the reviewer for bringing up simulator-based dataset construction methods. We also considered this approach alongside synthetic data generation early on, but ultimately decided against it for two reasons:
>
> **(1) Constrained to specific scenes:** Unlike image generation models, simulators (e.g., OmniGibson, AI2Thor, Habitat) offer a variety of environments but are inevitably constrained to specific scenes, limiting their coverage of the wide range of risks that can occur in real household environments.
>
> **(2) Requires manual effort:** Constructing risky scenarios under specific conditions inevitably requires manual scene configuration, which is significantly time-consuming. Furthermore, as we consider our automated synthetic data generation method to be a core contribution of our paper, we decided to exclude the simulator-based approach.
>
> We hope this addresses your question, and we would be happy to discuss any further concerns or questions you may have.

---

> > ### Author Rebuttal · Reviewer_a77M · 2026-04-04
> >
> > I appreciate the authors' effort, but the rebuttal does not change my judgment.

---

### Official Review · Reviewer_hjwN · 2026-03-13

**Soundness:** 3
**Presentation:** 3
**Significance:** 2
**Originality:** 3
**Overall Recommendation:** 4
**Confidence:** 2

**Summary:**

This paper proposes EMBGUARD, the first safety guardrail for embodied agents, which separates safety reasoning from the main agent policy. To train EMBGUARD, the authors introduce a three-stage dataset generation pipeline, resulting in the EMBHAZARD dataset. A manually curated evaluation benchmark, EMBGUARDTEST, is also introduced. A comprehensive evaluation of the guardrail is conducted on EMBGUARDTEST, a held-out set, and IS-Bench.

**Compliance With Llm Reviewing Policy:**

Affirmed.

**Final Justification:**

The paper introduces EMBGUARD with a new dataset and benchmark for physical risk assessment. My main concerns were about generalization beyond the predefined seven categories and the lack of experiments on larger backbones. While the second point remains a limitation, the rebuttal addressed my concern regarding Q1. Therefore, my final recommendation is weak accept.

**Key Questions For Authors:**

- The risks are categorized into seven categories. How well does EMBGUARD generalize to real-world scenarios beyond these categories?
- The EMBGUARD models with 2B and 4B parameters show relatively low Step Acc. and Precision. Have the authors explored scaling the guardrail model to larger backbones (e.g., Qwen3-VL-32B), and how would this affect performance?

**Limitations:**

yes

**Strengths And Weaknesses:**

Strengths:
- The proposed three-stage data generation pipeline is sound and rigorous, and two datasets are constructed using this pipeline.
- This appears to be the first guardrail model for embodied agents, which is novel.
- The insights regarding the systematic bias of current MLLMs are valuable.

Weaknesses:
- Please refer to the questions below.

---

> ### Author Rebuttal · Authors · 2026-03-31
>
> ## Initial Response to Reviewer hjwN
>
> We appreciate the reviewer for acknowledging the rigor of our data generation pipeline, recognizing the novelty of our EMBGuard model, and finding our insights valuable. We will address the questions you have raised below.
>
> ---
>
> # Q1: EMBGuard to Real-World Scenarios Beyond Our Categories
>
> We appreciate the reviewer for raising this question, as it gives us the opportunity to explain the rationale behind our categorization decisions. While performance cannot be guaranteed for inputs outside our 7 predefined categories, we believe such cases are unlikely to arise in practice, as **our categorization was designed to cover the scenarios where safety issues are most likely to occur in real-world embodied agents.** We will provide further reasoning for this below.
>
> **Our risk taxonomy was not simply derived from our own intuitions**, but was grounded in real-world incident reports and safety guidelines sourced from WHO ICD-11 (Chapter 23: External causes of morbidity or mortality) and the CPSC NEISS database, in order to comprehensively cover issues that arise in real-world industrial settings. From these sources, **we narrowed our scope to risks that can arise from external factors within indoor environments**, for the following reasons:
>
> (1) Embodied agents deployed outdoors and those deployed indoors are designed to handle fundamentally different tasks, and **our work targets problems specific to indoor settings.**
>
> (2) **Internal failure modes such as motion planning and collision avoidance fall within the responsibility of the embodied agent's policy**. Furthermore, since low-level motion operations must be executed repeatedly to carry out a high-level action, including these within the scope of the guardrail would significantly slow down policy execution. Therefore, **we designed our guardrail to intervene when a high-level planning decision fails to account for external risky factors, correcting such actions before execution.**
>
> (3) Additionally, as **we limited our input modalities to text and visual observations, we excluded risks that require non-visual sensing**, such as smell (e.g., gas leaks) and weight (e.g., exceeding a safe lifting threshold).
>
> Therefore, we believe our categorization sufficiently covers the risks that can arise in indoor environments. Should you have any further questions, please do not hesitate to ask.
>
> ---
>
> # Q2: Scalability of EMBGuard to Larger Backbone Models
>
> Thank you for suggesting this direction. We also believe that scaling to larger backbones such as Qwen3-VL-32B would likely bring further performance improvements. However, we did not train on models larger than 4B for two reasons: **(1) guardrail design considerations** and **(2) computational constraints.**
>
> First, regarding guardrail design considerations, **since we are designing a guardrail to assist the embodied agent's policy rather than the policy itself, fast inference speed is critical for the guardrail to effectively assist the embodied agent in real-time operation**. Therefore, scaling up the model size could hinder this requirement, which is why **we limited our training to the 2B and 4B model sizes**, despite their relatively lower performance compared to larger models. For this reason, although we had initially trained an 8B model, we ultimately decided to exclude it.
>
> Second, from a computational standpoint, to be honest, we do not have sufficient GPU resources to train LLMs size over 8B. We hope these two reasons sufficiently address your question.

---

> > ### Author Rebuttal · Reviewer_hjwN · 2026-04-03
> >
> > Thank you for the rebuttal. My concern regarding Q1 has been resolved. As for Q2, although the authors did not evaluate EMBGUARD on larger backbones, their rationale is reasonable.

---

### Official Review · Reviewer_cSv4 · 2026-03-13

**Soundness:** 2
**Presentation:** 3
**Significance:** 2
**Originality:** 2
**Overall Recommendation:** 4
**Confidence:** 4

**Summary:**

This paper proposes a safety guardrail for embodied agents that evaluates whether a candidate action is risky given the current visual scene and predicts the risk type and explains the underlying hazard. The central idea is to decouple physical safety reasoning from the agent policy and treat safety as a separate action-conditioned multimodal classification-and-explanation task. To support this, the paper introduces EMBHAZARD, a synthetic training dataset of 17K image-action pairs, and  a real-world benchmark of manually curated scenarios across seven physical risk categories. The experiment show that small specialized models fine-tuned on the proposed data can outperform comparable open models and approach much larger proprietary MLLMs on several metrics, while also reducing the over-conservative false-positive behavior that harms practical deployment.

**Compliance With Llm Reviewing Policy:**

Affirmed.

**Final Justification:**

the rebuttal partial solved my concerns on LLMaaJ evaluation and dataset curation. I would increase my score accordingly.

**Key Questions For Authors:**

1. EMBGUARD is implemented by supervised fine-tuning Qwen-3-VL models. As we noticed significant performance gap (even after finetuning) between 2B and 4B models, have the authors evaluated whether the gains are robust across different open-source backbones, or is the improvement sensitive to the choice of base model?

2. Could the authors provide more detail on how the synthetic EMBHAZARD training data are generated and filtered, and how similar they are to the real-world EMBGUARDTEST benchmark? In particular, it would be helpful to understand the overlap in object categories, action templates, scene compositions, and hazard patterns, since this directly affects how we should interpret the reported generalization results.

**Limitations:**

The authors discussed the broader impact of this work but I don't find discussions on limitations and potential negative impact.

**Strengths And Weaknesses:**

Strength:
* The paper is well motivated and the problem formulation is practically important. The paper correctly emphasizes that physical risk in embodied settings is action-conditioned: the same scene may be safe or unsafe depending on the proposed action.

* Methodology-wise, the three-stage pipeline (risk-driven scenario generation → compositional variation via scene graphs → image generation) is methodologically sound. Using scene graphs as an intermediate representation is a smart design choice that enables controlled manipulation of hazards and actions while preserving causal structure.

* The dataset contribution is valuable. EMBHAZARD and EMBGUARDTEST are carefully designed around both risky and benign cases, including harder settings such as hazard-present-but-safe-action and multi-hazard scenes with action-specific risk activation.

* The paper provides an important and empirical message: general-purpose MLLMs tend to be overly conservative in embodied safety settings, and this false-positive bias can severely hurt planning.

Weakness:
* The technical method is fairly simple. EMBGUARD is essentially a supervised fine-tuned Qwen-3-VL model trained on a newly constructed dataset.

* On the experiments, there are a few concerns here:

  1. Metrics: The hazard evaluation metric is somewhat soft. Hazard Accuracy is measured using GPT-4o as a judge on free-form text outputs, which is understandable but not ideal for a safety-critical task. The abstract and introduction cite reducing false positives as a key contribution, and this is supported by Figure 6. However, EMBGUARD achieves this partly by having lower recall than frontier models (e.g., EMBGUARD-4B recall on IS-Bench is 71.7%, vs. Gemini-2.5-Pro's 88.2%). The paper should more explicitly frame the precision-recall trade-off rather than presenting false-positive reduction as an unambiguous win.

  2. Experiment settings: The real-world benchmark contains only 189 samples, which is small for a safety-oriented evaluation. Another concern is the generalization/overfitting of the training data -  we notice the synthetic training data is somehow similar/correlated to the real world test set which may lead to some overfitting issues. IS-Bench evaluation (Section 5) partially validates this since it uses a completely separate simulator-based benchmark, which is why those numbers are arguably more meaningful for assessing real-world generalization. Notably, the performance gains there are more modest and the absolute numbers are harder to interpret.

---

> ### Author Rebuttal · Authors · 2026-03-31
>
> # W1: Concerns about technical simplicity
> We acknowledge that applying SFT on a newly constructed dataset may seem limited in technical novelty from a training perspective, and as we share this view, we did not emphasize the training contribution. Our primary focus lies in how to controllably generate synthetic data for guardrails in embodied agents, which, as the reviewer kindly noted in the Strengths, is where we believe our novelty lies.We hope this clarifies your concern about technical simplicity.
>
> # W2-1: Concerns related to Metrics
> Thank you for pointing out the concerns related to the metrics. We have separated your concerns into two parts and will address each of them individually below.
> ## W2-1 (a): LLM-as-a-Judge approach on Hazard accuracy
> ### **Due to space constraints, we kindly ask you to refer to Response to Reviewer 5mrG, W2 for our detailed response.**
> ## W2-1 (b): Precision-Recall Trade-off
> Initially, we also considered framing our results in terms of a precision-recall trade-off. However, based on our experimental insights, while consistent performance improvements are observed across trained models, the degree of improvement varies depending on the base model's capability. Nevertheless, since a reduction in false positives is consistently observed across both models, we chose to position our results around reducing false positives rather than framing them as a precision-recall trade-off, as this better reflects the overall and consistent improvements observed across both our synthetic and real-world datasets. We hope this addresses your concern and we are open to any further questions.
>
> # W2-2: Concerns related to experiment settings
> ## W2-2 (a): Real-world benchmark dataset size
> ### **Due to character limit constraints, we kindly refer the reviewer to our response to Reviewer a77M, W1 for our full response.**
>
> ## W2-2 (b): Concern about generalization/overfitting of the training data
> **We would like to first clarify a potentially confusing description in our EMBGuardTest construction process. It was manually constructed through a human labeling process, with images captured in real-life everyday environments.** For each predefined risk type, authors crafted scenarios, diversified them into four compositional variants, and applied a filtering process. Crucially, we ensured no overlap with the training set. Given this process, we believe it serves as a meaningful evaluation of real-world generalization. We will correct Section 3.2.3 and apologize for confusion.
>
> IS-Bench is, to our knowledge, the only benchmark evaluating safe planning for MLLM-based embodied agents. As a simulator-based benchmark, it relies on synthetic imagery, introducing a visual domain gap with our realistic training data (EMBHazard), which we believe accounts for the modest performance gains. We will add this to the Limitation Section.
>
> # Q1: Performance gain robustness
>
> To answer your question about performance robustness across different backbone models, we present the EMBGuardTest results trained with various backbone models in the table below.
>
> |Backbone|Potential Risk ($\Delta$)|Risk Type ($\Delta$)|Hazard ($\Delta$)|
> |-|-|-|-|
> |InternVL3.5-1B|37.6 (+23.2)|24.6 (+14.2)|10.5 (+2.8)|
> |InternVL3.5-2B|56.1 (+8.0)|37.2 (+22.8)|10.5 (+3.4)|
> |Gemma-3-4B|57.3 (+19.3)|35.8 (+11.7)|9.9 (+2.2)|
>
> In addition to the results shown in the table, we would like to share our experimental insights. **Training on our dataset consistently leads to performance improvements across all models.** However, as you have pointed out, the magnitude of these gains varies. **We believe this variation is driven more by the inherent capabilities of each backbone model rather than the model family it belongs to.** We will include this analysis in Appendix A.
>
> # Q2: Detailed analysis of EMBHazard & EMBGuardTest
>
> ## Q2 (a): EMBHazard construction process
>
> To ensure dataset quality, we manually constructed few-shot prompts and applied LLM-based quality filtering across all text-based generation stages to verify the quality. **After submission, we additionally applied a VQA-based filtering step using GPT-5.1, refining the dataset to 15.1K samples, which has been made publicly available.** We will add this in Section 3.2.3.
>
> ## Q2 (b): Train-Test overlap analysis
>
> **Object Overlap.** Over 65% of objects in EMBGuardTest appear 0–10 times in EMBHazard, suggesting largely distinct object configurations. The small fraction appearing 100+ times (3.33%) is attributable to inherently common household objects.
>
> **Action Template.** EMBGuardTest shows lower exact match (30.38%) and cosine similarity (88.61%) with EMBHazard, reflecting its independent and manual curation.
>
> **Hazard Pattern Overlap.** At a strict threshold of 0.85, only 9.02% of EMBGuardTest samples show high similarity to training scenarios, indicating that the vast majority represent novel hazard patterns unseen during training.
>
> We will include the detailed analysis results in Appendix A.

---

> > ### Author Rebuttal · Reviewer_cSv4 · 2026-04-04
> >
> > I appreciate the authors' effort and the rebuttal partial solved my concerns on LLMaaJ evaluation and dataset curation. I would increase my score accordingly.

---

### Decision · Program_Chairs · 2026-04-30

**Decision:**

Accept (regular)

**Comment:**

In this paper, the authors propose EMBGUARD, a multimodal safety guardrail for embodied agents that evaluates whether a candidate action in the physical world poses a safety risk given the current visual scene, predicts the risk category, and explains the underlying hazard. The central design choice is to decouple physical safety reasoning from the agent's main policy, treating safety as a separate action-conditioned classification task. The authors introduce EMBHAZARD, a ~15K synthetic training dataset built via a three-stage pipeline (risk-driven scenario generation, compositional variation through scene graphs, and image generation) covering seven physical risk categories grounded in official sources, and EMBGUARDTEST, a real-world benchmark expanded from 189 to 329 instances during rebuttal. Compact 2B/4B Qwen-3-VL models fine-tuned on EMBHAZARD match or approach larger proprietary MLLMs while reducing the over-conservative false-positive bias observed in general-purpose models.

Post-rebuttal reviews ranged over three weak accepts (R-cSv4, R-hjwN, R-a77M) and one weak reject (R-5mrG). Strengths cited across reviewers: the action-conditioned framing of physical risk, the compositional dataset design via scene graphs, and the false-positive-reduction result as a practical deployment signal. Concerns covered the small real-world benchmark size and potential train-test overlap (R-cSv4, R-a77M, R-5mrG); LLM-as-judge reliability for hazard accuracy (R-cSv4, R-5mrG); backbone scalability and robustness (R-cSv4, R-hjwN); generalization beyond the 7 predefined categories (R-hjwN); image-generator sensitivity (R-a77M); partial observability from per-step evaluation ignoring sequential context (R-5mrG); and limited applicability to continuous-control VLAs (R-5mrG).

In the rebuttal, the authors expanded EMBGUARDTEST to 329 manually-curated instances with no training overlap and re-ran all experiments, reporting high synthetic-to-real correlation (r ~0.96 on Potential Risk); a human-judge agreement study on GPT-4o hazard judgments yielding kappa = 0.90 against majority-voted annotators; backbone-robustness experiments across InternVL3.5, Qwen-3-VL, and Gemma-3 at 1B–4B scales; a train-test overlap analysis showing limited object/action/hazard pattern overlap; clarified rationale for the 7 risk categories (grounded in safety incident taxonomies); and explicit limitations around partial observability, visual sensor coverage, and continuous-control applicability. R-cSv4 indicated they would raise their score; R-hjwN marked concerns fully resolved; R-a77M kept their score without articulating remaining objections; R-5mrG acknowledged two of the initially pointed out weaknesses were resolved but held weak reject on residual partial-observability concerns, which the authors framed as a deliberate real-time-deployment tradeoff and added to the limitations section.

The core contribution of an action-conditioned safety guardrail with a compositional synthetic dataset pipeline, an expanded real-world benchmark, and empirical results aobut false-positive bias in general-purpose MLLMs is practically valuable and well-motivated. The rebuttal strengthened the empirical base (expanded benchmark, human-judge validation, backbone-robustness, overlap analysis) and added the main methodological concerns as acknowledged limitations. Three of four reviewers support acceptance, and R-5mrG's residual partial-observability concern is more a legitimate future-work direction than a fundamental flaw given the real-time-deployment framing. I recommend acceptance, and encourage the authors to integrate the expanded EMBGUARDTEST results, human-judge agreement study, backbone-robustness analysis, and the added limitations section into the main text.